# Diurnal Regulation of Leaf Photosynthesis Is Related to Leaf-Age-Dependent Changes in Assimilate Accumulation in *Camellia oleifera*

**DOI:** 10.3390/plants12112161

**Published:** 2023-05-30

**Authors:** Jinshun Zhang, Lingyun Zhang, Qi Wang, Jiali Liu, Yongjiang Sun

**Affiliations:** Key Laboratory of Forest Silviculture and Conservation of the Ministry of Education, The College of Forestry, Beijing Forestry University, 35 Qinghua East Road, Beijing 100083, China; 15650455250@163.com (J.Z.);

**Keywords:** *Camellia oleifera*, leaf age, high light, assimilate accumulation

## Abstract

In order to clarify the mechanism of diurnal changes in photosynthesis of leaves of different leaf ages in *Camellia oleifera*, current-year leaves (CLs) and annual leaves (ALs) were used as the test materials to analyze the diurnal changes in photosynthetic parameters, assimilate contents and enzyme activities, as well as structural differences and expression levels of sugar transport regulating genes. The rate of net photosynthesis in CLs and ALs was highest in the morning. During the day, there was a decrease in the CO_2_ assimilation rate, and this decrease was greater in ALs than in CLs at midday. The maximal efficiency of photosystem II (PSII) photochemistry (*F*_v_/*F*_m_) showed a decreasing trend as the sunlight intensity increased, but no significant difference between CLs and ALs was found. Compared with CLs, ALs showed a greater decrease in the carbon export rate at midday and the levels of sugars and starch increased significantly in ALs, accompanied by higher enzyme activity of sucrose synthetase and ADP-glucose pyrophosphorylase. In addition, compared with CLs, ALs had a larger leaf vein area and higher leaf vein density, as well as higher expression levels of sugar transport regulating genes during the day. It is concluded that the excessive accumulation of assimilate is an important factor contributing to the midday depression of photosynthesis in *Camellia oleifera* annual leaves on a sunny day. Sugar transporters may play an important regulatory role in excessive accumulation of assimilate in leaves.

## 1. Introduction

Plants fix CO_2_ into carbohydrates through photosynthesis; sucrose is the major photoassimilate in many plant species, which is transported to non-photosynthesizing organs, providing substrates for the growth and reproduction [1]. Therefore, increasing the rates of leaf photosynthesis and assimilate conversion is required to improve yield potential. However, plants do not always conduct photosynthesis under optimal conditions; various environmental stresses, such as high light, high air temperature and high vapor pressure difference (VPD) can cause strong midday depression of photosynthesis, known as the phenomenon of a photosynthetic ‘lunch break’ [2]. The reasons for the phenomenon of the photosynthetic ‘lunch break’ are complex, mainly including stomatal limitations caused by stomatal closure and various non-stomatal limitations [3]; the biochemical reaction ability of leaves is the main limiting factor for photosynthesis among non-stomatal limitations [4,5].

Light conditions are the primary conditions for photosynthesis. However, high light can lead to an excess of light energy required for photosynthesis, resulting in a decrease in photosynthesis [6,7,8]. High light can also affect the transportation of photosynthetic assimilates, leading to an increase in the accumulation of assimilates in the source leaves [9,10]. Due to the limited ability of leaves to store photosynthates, on average, the remaining newly fixed carbon is either consumed in local respiration or exported within 24 h [11]. If the assimilates cannot be effectively exported from the source leaves, photosynthesis will be severely inhibited [12,13,14]. The transportation of assimilates in leaves is closely related to the structure of the phloem and the density of leaf veins [15]. In addition, sugar transporters (mainly SUTs and SWEETs) play an important role in the phloem loading of sugars [16,17]. As leaves grow, their ability for photosynthetic carbon assimilation and photoassimilate transport increases [18,19]. Correspondingly, it is believed that regulation of leaf photosynthesis is strongly dependent on leaf age and that assimilate accumulation in leaves of different ages may exhibit differential sensitivity to high light [20]. Therefore, it is necessary to clarify the mechanism of diurnal changes in photosynthesis and photoassimilate transport characteristics of leaves of different leaf ages in natural conditions.

*Camellia oleifera* Abel. is one of the four major woody oil trees in the world [21] but suffers from low yields in most cultivation areas of China. The raw material for oil synthesis in seeds comes from the decomposition of sugar, whereas sugar mainly comes from the photosynthesis of leaves [22]. As an evergreen tree species, there are both current-year leaves and leaves over a year old in the crown; however, fewer studies have been performed on the relationship between diurnal regulation of photosynthesis and the assimilate transport characteristics of leaves of different leaf ages. In the present study, an integrative analysis of diurnal changes in photosynthetic parameters, assimilate contents and enzyme activities related to carbon metabolism was performed in current-year leaves (ALs) and annual leaves (CLs). More importantly, the structure of leaf veins and the expression of genes directly related to assimilate transport were studied. The objective of this study was to clarify the mechanism of diurnal changes in photosynthesis and assimilate transport characteristics of leaves of different leaf ages in *Camellia oleifera*.

## 2. Results

### 2.1. Diurnal Variation in the Gas Exchange of Leaves of Different Leaf Ages in Camellia oleifera

The diurnal variation characteristics of photosynthetic photon flux density (PPFD) and air temperature (*T*_a_) during measurement are shown in Figure 1. The maximum PPFD and *T*_a_ recorded were around 1500 μmol·m^−2^·s^−1^ and 37 °C at midday, respectively (Figure 1A). As shown in Figure 2B, the results showed that light response curves in ALs and CLs were of a similar tendency. The light saturation points (LSP) in ALs and CLs were much lower than the intensity of direct sunlight during most of the day. When PPFD exceeded LSP, the net photosynthetic rate (*P*_n_) of ALs was higher than that of CLs. In fact, the PPFD at 10:00 h was recorded as around 1300 μmol·m^−2^·s^−1^, which had already exceeded the LSP of *Camellia oleifera*.

The difference in chlorophyll content between ALs and CLs is shown in Figure 2A. The total chlorophyll, chlorophyll a, and chlorophyll b contents in ALs were significantly higher than those in CLs (*p* < 0.05), thus indicating that the light absorption and transformation ability in ALs might be superior to that in CLs.

*P*_n_ reached peak values at 10:00 h (10.26 μmol·m^−2^·s^−1^ in ALs and 6.83 μmol·m^−2^·s^−1^ in CLs) and then sharply decreased. From 10:00 h to 14:00 h, the drop percentages for *P*_n_ in ALs and CLs were 69.5% and 28.1%, respectively, and the decline was faster in ALs than that in CLs (Figure 2B). There was then some recovery from 14:00 h to 16:00 h (39.0% and 12.2%, respectively). The diurnal variation in stomatal conductance (*G*_s_) was similar to the *P*_n_ (Figure 2C). The *G*_s_ in ALs and CLs reached the maximal values of 95.7 mmol·m^−2^·s^−1^ at 10:00 h and 72.1 mmol·m^−2^·s^−1^ at 10:00 h followed by some decreases from 10:00 h to 14:00 h and slight increases from 14:00 h to 16:00 h. In contrast to *P*_n_, the intercellular CO_2_ concentration (*C*_i_) in ALs and CLs increased by 27.0% and 13.1% from 8:00 h to 12:00 h, respectively, and basically remained constant in the afternoon despite a slight decrease at 14:00 h in CLs (Figure 2D). During the period of the sharp decline in *P*_n_ in ALs from 10:00 h to 14:00 h, *G*_s_ also continued to decline, whereas *C*_i_ showed an upward trend, indicating that the decline in *P*_n_ in ALs was mainly caused by non-stomatal limiting factors. However, the trend of *G*_s_ and *C*_i_ in CLs was consistent, indicating that the decline in *P*_n_ was mainly caused by stomatal limiting factors.

### 2.2. Changes in the Chlorophyll Fluorescence Parameters of Leaves of Different Leaf Ages in Camellia oleifera

In order to explore the difference in the photosynthetic mechanism activity between the ALs and CLs of *Camellia oleifera* and analyze the photoinhibition levels, the chlorophyll fluorescence parameters of ALs and CLs were measured (Figure 3). The diurnal variation trend of the maximal efficiency of photosystem II (PSII) photochemistry (*F*_v_/*F*_m_) of ALs and CLs was consistent, which showed a decreasing trend as the sunlight intensity increased. From 8:00 h to 14:00 h, the drop percentages of *F*_v_/*F*_m_ in ALs and CLs were 7.2% and 7.9%, respectively. However, there was no significant difference between CLs and ALs (Figure 3A). At noon, the actual photochemical efficiency (Y(II)) significantly decreased in ALs and CLs. From 8:00 h to 14:00 h, the drop percentages of Y(II) in ALs and CLs were 69.2% and 55.7%, respectively. Meanwhile, the values of Y(NPQ) largely increased (93.2% and 48.6%, respectively) to dissipate excess absorbed light energy, accompanied by a lower quantum yield of non-regulated energy dissipation (Y(NO)), indicating that the extent of the PSII photoinhibition trends in ALs and CLs coincide.

### 2.3. Dark Respiration and the Carbon Export Rate of Leaves of Different Leaf Ages in Camellia oleifera

In order to investigate the respiratory consumption of photoassimilates in ALs and CLs, the dark respiration rates (*R*_d_) in ALs and CLs were measured. The results showed that the dark respiration rates (*R*_d_) in ALs and CLs at midday were the highest during the day. Compared with CLs, ALs showed a smaller increase in the level of *R*_d_ from 10:00 h to 16:00 h (46.6% and 167.7%, respectively), and the *R*_d_ of CLs was significantly higher than that of ALs (Figure 4A).

The carbon export rates of ALs and CLs were also measured, which could reflect the assimilation transport capacity of the leaves. The results showed that the carbon export rates of CLs and ALs were highest in the morning, which was followed by a gradual decrease across the day. However, this reduction was greater in ALs than that in CLs (Figure 4B). From 10:00 h to 14:00 h, the drop percentages for the carbon export rate in ALs and CLs were 38.9% and 25.5%, respectively.

### 2.4. Assimilate Metabolism Differences of Leaves of Different Leaf Ages in Camellia oleifera

In order to examine whether the regulation of leaf photosynthesis is associated with the levels of assimilate accumulation, we investigated the changes in the sugar and starch contents and related major metabolic enzyme activities of ALs and CLs. From 10:00 h to 14:00 h, the sucrose content of ALs and CLs increased (15.9% and 36.3%, respectively), and this increase was greater in ALs than in CLs (Figure 5A). The content of soluble sugar also increased by 8.0% at midday in ALs but showed a decrease of 14.9% in CLs (Figure 5B). The changes in starch levels were similar to those of soluble sugar, with an increase of 61.2% in ALs and a decrease of 40.0% in CLs. In addition, the starch content of CLs significantly increased at night (Figure 5C).

The activity of sucrose synthase (synthesis direction) was greater in ALs than in CLs throughout the day (Figure 5D), and from 14:00 h to 22:00 h, the percentage increase in ALs was greater than in CLs (105.0% and 13.9%, respectively). The activity of sucrose phosphate synthetase in ALs and CLs remained essentially unchanged (Figure 5E). The activity of ADP-glucose pyrophosphorylase, the key enzyme for starch synthesis, was significantly higher in ALs than in CLs at midday. However, from 14:00 h to 22:00 h, the activity of ADP-glucose pyrophosphorylase in ALs was significantly reduced (88.7%), whereas the activity in CLs increased by 50.5% (Figure 5F).

### 2.5. Structural Differences of Leaves of Different Leaf Ages in Camellia oleifera

To further explore the factors affecting the accumulation of assimilates, the leaf vein structures of ALs and CLs was observed, which was the basis for the outward transport of assimilates. The leaf vein network skeleton of the leaves was analyzed and classified using LEAF GUI software, whereas the leaf venation information of leaves was extracted (Figure 6A). The results showed that the number of leaf veins, total length of leaf veins and leaf vein density values in ALs were all significantly higher than the corresponding values in CLs (Figure 6C).

In addition, we also observed the anatomical structures of ALs and CLs (Figure 6B). The results showed that the main vein thickness, main vein area and minor vein area values for ALs were all significantly larger than the corresponding values in CLs (Figure 6C). Our results indicate that the leaf vein structure of ALs was more developed than that of CLs, which would facilitate the transport of assimilates.

### 2.6. Gene Expression Differences in Leaves of Different Leaf Ages in Camellia oleifera

Sugar transporters play an important role in the transport of sugar. In previous studies, several genes related to sugar transport, including *CoSUT1*, *CoSUT2-1*, *CoSUT2-2* and *CoSWEET12*, were found to be differentially expressed in leaves of different leaf ages in *Camellia oleifera* [23]. The expression levels of the abovementioned differential genes were detected in this experiment (Figure 7). During the day, the expression levels of *CoSUT1*, *CoSUT2-1*, *CoSUT2-2* and *CoSWEET12* in ALs were all highest at 10:00 h and were all higher than those in CLs at this time. At 14:00 h, the expression levels of *CoSUT1*, *CoSUT2-1* and *CoSWEET12* were higher in ALs than in CLs. In addition, the expression level of *CoSUT2-1* in CLs was highest at 22:00 h and significantly higher than that in ALs. The results indicate that the expression levels of several genes related to sugar transport were higher in ALs in daytime, which would promote the transport of assimilates.

## 3. Discussion

### 3.1. Analysis of the Differences in the Photosynthetic ‘Lunch Break’ of Leaves of Different Leaf Ages

Leaves are the major organs for photosynthetic carbon assimilation. As an evergreen tree species, *Camellia oleifera* has a leaf life span of more than one year. The amount chlorophyll in the leaf is an important indicator of the photosynthetic capacity and leaf development of *Camellia oleifera* tissue [24]. In our study, the chlorophyll content of ALs was higher than that in CLs, with a higher capacity for gas exchange in the morning. However, *P*_n_ values for ALs decreased more significantly at noon, accompanied by a decrease in *G*_s_ and an increase in *C*_i_, indicating that more severe non-stomatal limitations occurred in ALs, which in fact is due to the metabolic constraints in leaves [25].

In C3 plants, midday depression of photosynthesis is a common phenomenon [2]. In our study, a light-saturation point of about 800 μmol·m^−2^·s^−1^ was found in ALs and CLs. However, the PPFD easily exceeded 1200 μmol·m^−2^·s^−1^ at 10:00 h. In the meantime, there was a decrease in *F*_v_/*F*_m_ and activation of Y(NPQ), indicating that photoinhibition of photosynthesis occurred in both ALs and CLs. However, no significant difference between CLs and ALs was found, indicating that neither photoinhibition nor photoprotection differed significantly between CLs and ALs at midday [26], as reported previously in soybeans [27].

In addition to photoinhibition caused by high light, the feedback regulation of the accumulation of assimilates in leaves can also cause the midday depression of photosynthesis [28,29]. In our study, we measured the accumulation of assimilates in ALs and CLs, including the content of soluble sugar and starch, as well as the activities of related major metabolic enzymes. The results showed that the contents of sucrose, total soluble sugar and starch in ALs were significantly higher than those in CLs at midday, indicating that ALs accumulated more assimilates through conducting a higher level of photosynthesis in the morning. Our results also showed that ALs had a higher synthetic activity for sucrose and starch at midday. In studies of other species, it has also been proven that the excessive accumulation of carbohydrates inhibits carbon assimilation and ultimately photosynthesis [30,31,32]. It has been suggested that high light can also affect the transportation of assimilates [10,33,34], thus leading to feedback inhibition of assimilate accumulation. In our study, *Camellia oleifera* seedlings were subjected to photoinhibition caused by high light at midday. Meanwhile, due to the accumulation of more assimilates in the morning, photosynthesis in ALs was more inhibited at midday. These results suggested that more severe midday depression of photosynthesis in ALs might be related to the excessive accumulation of photoassimilates.

### 3.2. Regulation of the Assimilate Transport in Leaves of Different Leaf Ages

As mentioned above, effective transportation of leaf assimilates is crucial for alleviating the feedback inhibition caused by assimilate accumulation and improving photosynthesis. Previous studies have shown that the CO_2_ fixation capacity of leaves is positively correlated with the transport level of assimilates [35,36]. Lanoue et al. [37] quantified the diurnal pattern of carbon export through ^14^CO_2_ labeling combined with the measurement of carbon sequestration and respiration rate and found that there was a very high correlation between photosynthesis and carbon export. Up to now, the carbon balance of leaves has been applied to evaluate the transport dynamics of assimilates in several plants [11,38,39,40]. In our study, we quantified the carbon export rates of *Camellia oleifera* leaves throughout the day by calculating the organic carbon content and net photosynthetic rate. The rates of carbon export in leaves were consistent with the net photosynthetic rate, as previous research has reported [11,40]. During the day, there was a decrease in the carbon export rate at midday, and this decrease was greater in ALs than in CLs. It is worth noting that compared with CLs, the *P*_n_ in ALs decreased more significantly at midday, accompanied by a higher level of carbon export rate. Given that the *P*_n_ was low at this time, with a high level of starch accumulation, we surmise that the assimilates generated in the morning would be exported at noon.

As the essential constituents of the leaves, the leaf veins play a crucial role in transporting assimilates [41]. A recent study found that *Camellia oleifera* mainly adopts a passive phloem loading mode [23]. The efficiency of passive loading is directly related to leaf vein and intercellular filament density and is easily affected by the concentration of carbohydrates at the source end [16,17]. In our study, compared with CLs, ALs showed a higher vein density, larger cross-sectional areas of main and minor veins and a larger total length of veins, indicating that the more developed leaf vein structure of ALs was more conducive to carbon export. However, leaves cannot increase their leaf vein density and intercellular filament density in a short period of time [42]. The present study showed that the expression levels of several sugar transporter regulatory genes in ALs were significantly higher than those in CLs during the day. This indicated that sugar transporters may play a greater role in ALs when the accumulation of assimilates occurs in ALs. This was consistent with the increased expression of sugar transporters in the mature and/or senescent leaves found in other studies [23,43,44]. Our results showed that when the accumulation level of leaf assimilates was high, the export of assimilates may be regulated by enhancing the expression of sugar transporters, which allows plants to consume the carbon stored in the leaves [17,45] and helps plants to adapt to changes in the assimilation levels in a short timeframe.

The present study showed that the midday depression of *P_n_* in *Camellia oleifera* seedlings occurred on a sunny day. In addition to the inactivation of photosystem activities caused by excessive excitation energy under high light, in our study, the age-dependent accumulation of assimilates was suggested as another mechanism for diurnal regulation of leaf photosynthesis in *Camellia oleifera* (Figure 8). Similar results were obtained in some shade-tolerant tree species, which are easily affected by high light [10,33,34]. Therefore, it is suggested that appropriate measures for shade enclosure should be adopted to mitigate the risks of high light during *Camellia oleifera* seedling cultivation.

## 4. Materials and Methods

### 4.1. Plant Materials

Two-year-old *Camellia oleifera* ‘Cenruan 3’ grafted seedlings, grown in a greenhouse with an average temperature of 30 °C and a daily maximum light intensity of 800 μmol·m^−2^·s^−1^, were used as the experimental materials in summer, about one month after the new leaves fully unfolded.

As shown in Figure 9, the ages of the branches and leaves were determined by the presence of bud scale marks. The leaves that grew on the new shoots and annual branches were regarded as current-year leaves (CLs) and annual leaves (ALs), respectively. A total of 27 seedlings with the same growth trend were selected and tested during consecutive sunny days for a total of three days. At each time point, a total of 9 CLs and 9 ALs in the same position were taken, then immediately packaged in aluminum foil, frozen in liquid nitrogen and stored at −80 °C.

### 4.2. Determination of Chlorophyll Content

Chlorophyll content was determined by the ethanol/acetone extraction method [46]. The leaves were washed with distilled water. Each sample was weighed as 0.1 g, ground fully, and then transferred into a 25 mL test tube. Approximately 10 mL of extraction solution was added into each tube; then, the tubes were sealed and stored closed to light at room temperature for 24 h. They were then examined through colorimetric analysis. The optical density was measured at wavelengths of 663 nm and 645 nm. The chlorophyll content was calculated by the Lambert–Beer law.

### 4.3. Determination of Diurnal Gas Exchange Parameters and the Carbon Export Rate

The diurnal variation in photosynthesis and the dark respiration rate of the leaves were observed by an Li-6400XT photosynthetic measurement system (Li-Cor. NE, USA). From 6:00 h to 18:00 h on a clear, cloud-free day, the net photosynthetic rate (*P*_n_), stomatal conductance (*G*_s_) and intercellular CO_2_ concentration (*C*_i_) were measured every two hours; the environmental factors were recorded, including the photon flux density (PPFD) and the air temperature (*T*_a_) (Figure 1A). The measurement of the leaf dark respiration rate referred to the method of Reich et al. [47]. After completely covering the leaf chamber with a black cloth for 10 min, the apparent photosynthetic rate at a PPFD of 0 μmol·m^−2^·s^−1^ was used as the dark respiration rate (*R*_d_).

The light response process of the photosynthetic rate of the leaves was measured from 9:00 h to 10:00 h on clear, cloud-free day. The built-in red and blue light sources were used, and PPFD was set to 1200, 800, 500, 200, 150, 100, 50 and 0 μmol·m^−2^·s^−1^. The modified model of the rectangular hyperbola was selected to fit the light response curve [48,49,50]. After measuring the photosynthetic rate and light response of *Camellia oleifera* leaves, it was calculated that the saturation light intensities in ALs and CLs were 797.03 μmol·m^−2^·s^−1^ and 812.07 μmol·m^−2^·s^−1^, respectively (Figure 1B).

Determination of the carbon export rate was based on the method of Gersony et al. [11], with slight changes. Net photosynthesis was defined as the net carbon flow rate with consideration of carbon loss via respiration in leaves. Therefore, the carbon export rate was calculated using a mass balance approach in order to reflect the transportation level of photoassimilates. The carbon export rate was calculated according to the difference between the content of organic carbon fixed by photosynthesis and the content of organic carbon accumulated in the leaves; the content of organic carbon was measured by a total organic carbon analyzer. The calculation formula is:Export = *A*_n_ − (*C*_n_/*T*_n_)
where export is the carbon export rate, *A*_n_ is the average value of the net assimilation 30 min before and after a specific time point and converted into the rate of organic carbon produced (μmol C·m^−2^·s^−1^), *C*_n_ is the difference in the organic carbon content of the leaves before and after the specific time point and is standardized by the leaf sampling area, and *T*_n_ is the sampling interval before and after the specific time point (1 h).

### 4.4. Determination of Diurnal Chlorophyll Fluorescence Parameters

The chlorophyll fluorescence parameters of the target leaves were measured using PAM-2500 (Walz, Effeltrich, Germany). After 20 min of dark adaptation, the initial fluorescence *F*_o_ and maximum fluorescence *F*_m_ under the dark adaptation of the leaves were measured. Then, the steady-state fluorescence *F*_s_ and maximum fluorescence *F*_m_′ natural under light were measured after sufficient light activation. According to Kramer et al. [51], the fluorescence parameters were calculated using the following formula:

Maximum photochemical efficiency of PSII:*F*_v_/*F*_m_ = (*F*_m_ − *F*_o_)/*F*_m_

Actual photochemical efficiency of PSII; under light:YII = (*F*_m_′ − *F*_s_)/*F*_m_′

Quantum yield of unregulated energy dissipation:
Y(NO) = *F*_s_/*F*_m_

Quantum yield by adjusting energy dissipation:Y(NPQ) = 1 − YII − Y(NO)

### 4.5. Determination of Soluble Sugar and Starch Content and Related Metabolic Enzyme Activity

The determination of total soluble sugar and starch content was carried out using a plant soluble sugar content detection kit (Solarbio, Beijing, China) and a starch content analysis kit (Solarbio, Beijing, China). The determination of sucrose content, sucrose synthase activity (synthesis direction) and sucrose phosphate synthase activity was carried out with reference to the methods in the experimental instructions [46,52]. The determination of ADPG pyrophosphorylase activity was slightly modified according to the method of Kulichikhin et al. [53].

### 4.6. Extraction of Leaf Venation Information and Observation of Leaf Structure

The extraction of leaf venation information referred to the method of Price et al. [54]. LEAF GUI software (http://www.leafgui.org/ (accessed on 10 September 2020)) was used to extract leaf vein venation information, and different levels of leaf veins were divided according to the method of Ma et al. [55]. Parameters such as the number of small segments of leaf veins, the total length of the leaf veins, the leaf area and the calculated leaf vein density (total length of leaf veins/leaf area) were obtained.

The anatomical structure of the leaves was determined by the conventional paraffin section method [56]. Leaves of the two different ages were taken during the above determination period and washed with clean water and dried with absorbent paper. Tissue blocks from the main veins and small veins in the middle of the leaves were cut, and the cut samples were fixed in FAA-fixed solution (70% ethanol, glacial acetic acid and formaldehyde: volume ratio 18:1:1). The samples were dehydrated with alcohol and xylene, made transparent, soaked in wax and then cut into slices. After dewaxing, the slices were dyed with saffron and solid green and sealed with neutral gum. The anatomical structure of the leaves was observed and photographed with a DM2500 biological microscope (Leica, Wetzlar, Germany), and parameters such as the thickness and area of the veins were measured with Image J software 1.8.0.

### 4.7. Analysis of Gene Expression Levels

Gene selection was carried out through the transcriptome analysis of *Camellia oleifera* leaves of different leaf ages conducted by Yang et al. [23] in our laboratory. The differential genes related to sugar transport in ALs and CLs of *Camellia oleifera* were selected for analysis. Oligo 7 was used to design the qRT-PCR primers (Table 1), and the primers were synthetized by Beijing Ruibo Xingke Biotechnology Co., Ltd. The samples were ground with liquid nitrogen, and total RNA was extracted with an OminiPlant RNA Kit (CoWin Biotechnology, Beijing, China). A StepOne Plus Real-Time PCR System (ABI, Vernon, CA, USA) was used to synthesize the first strand of cDNA as a template. The StepOne Plus Real-Time PCR System was used for the PCR reaction using 2 × Fast SYBR Green qPCR Master Mix (Thermo Fisher Scientific, Waltham, MA, USA). *CoACTR3* was selected as the internal reference gene [24]. The 2^−ΔΔCT^ method was used to analyze the data.

### 4.8. Data Analysis

Excel and SPSS Statistics 17 were used for data statistics and analysis. Statistical analyses were carried out using one-way ANOVA and the LSD (L) multiple comparisons test (*p* < 0.05) in SPSS 17. SigmaPlot 10.0 was used for image rendering.

## 5. Conclusions

In this study, we compared the structure, photosynthetic physiological characteristics and assimilate transport capacity of leaves of different ages in *Camellia oleifera* seedlings. The results showed that diurnal regulation of leaf photosynthesis showed a leaf-age-dependent change in assimilate accumulation. The midday depression of photosynthesis in *Camellia oleifera* annual leaves on a sunny day was associated with the excessive accumulation of assimilate. The well-developed leaf vein structure and the increased expression level of sugar transport regulating genes during the accumulation of assimilates in annual leaves were likely beneficial for the transport of assimilates.

## Figures and Tables

**Figure 1 plants-12-02161-f001:**
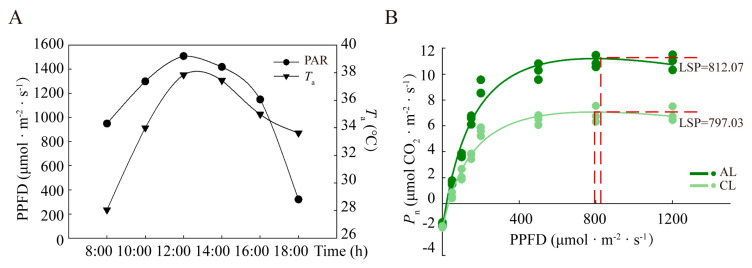
The diurnal variation of environmental factors and the photosynthetic light response curve. (**A**) Diurnal changes in photosynthetic photon flux density (PPFD) and air temperature (*T*_a_), (**B**) Light response curves of net photosynthetic rate (*P*_n_). The light response curves were measured from 9:00 h to 10:00 h on a clear, cloud-free day. Data were fit to the modified model of rectangular hyperbola. LSP: light saturation point; ALs: annual leaves; CLs: current-year leaves.

**Figure 2 plants-12-02161-f002:**
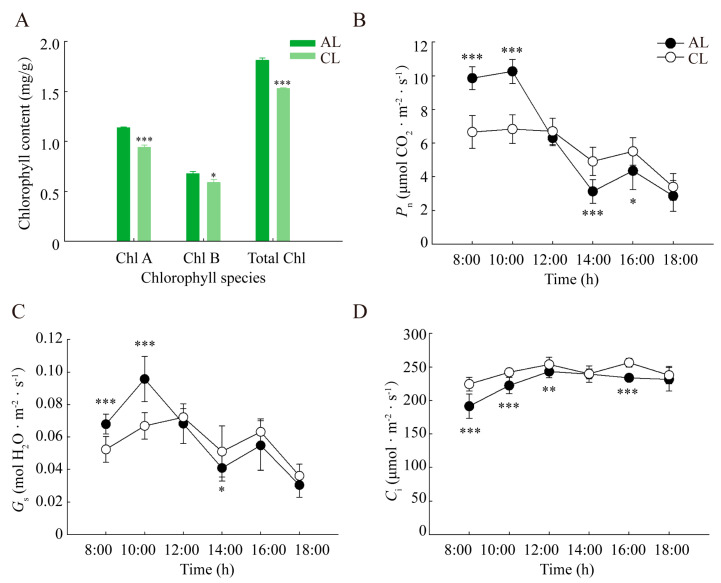
The diurnal variation of (**A**) chlorophyll content, (**B**) the net photosynthetic rate (*P*_n_), (**C**) stomatal conductance (*G*_s_) and (**D**) intercellular CO_2_ concentration (*C*_i_) of ALs and CLs in *Camellia oleifera*. ALs: annual leaves; CLs: current-year leaves. * *p* < 0.05; ** *p* < 0.01; *** *p* < 0.001.

**Figure 3 plants-12-02161-f003:**
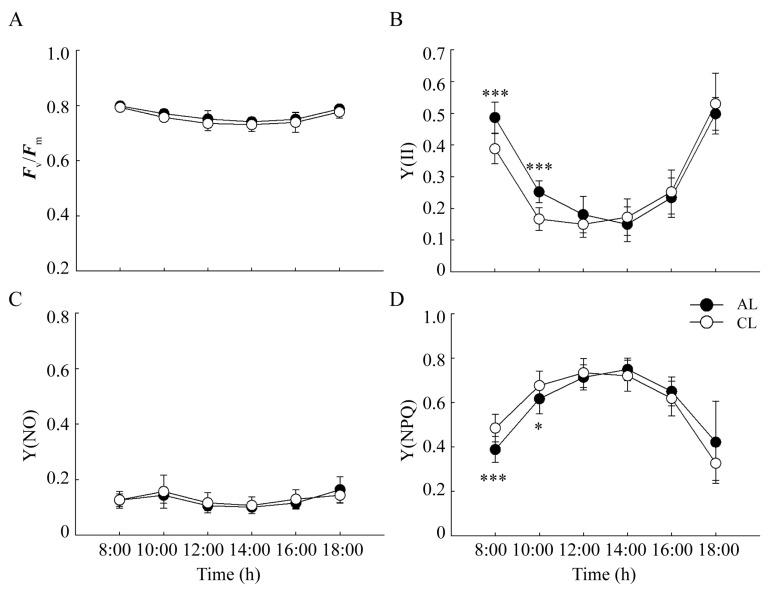
The diurnal variation in chlorophyll fluorescence parameters in ALs and CLs of *Camellia oleifera.* (**A**) Maximum quantum yield (*F*_v_/*F*_m_), (**B**) effective PSII quantum yield (Y(II)), (**C**) quantum yield of non-regulated energy dissipation (Y(NO)), (**D**) quantum yield of regulated energy dissipation [Y(NPQ)]. ALs: annual leaves; CLs: current-year leaves. * *p* < 0.05; *** *p* < 0.001.

**Figure 4 plants-12-02161-f004:**
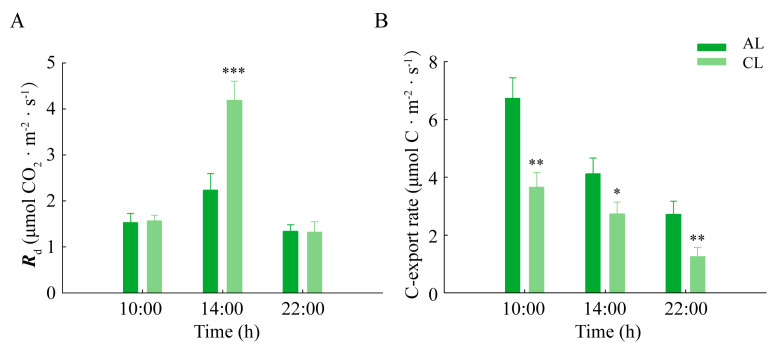
The dark respiration rate (**A**) and carbon export rate (**B**) in ALs and CLs of *Camellia oleifera.* The dark respiration rate (*R*_d_) was the apparent photosynthetic rate at the PPFD of 0 μmol·m^−2^·s^−1^. The carbon export rate was calculated according to the difference between the content of organic carbon fixed by photosynthesis and the content of organic carbon accumulated in the leaves. ALs: annual leaves; CLs: current-year leaves. * *p* < 0.05; ** *p* < 0.01; *** *p* < 0.001.

**Figure 5 plants-12-02161-f005:**
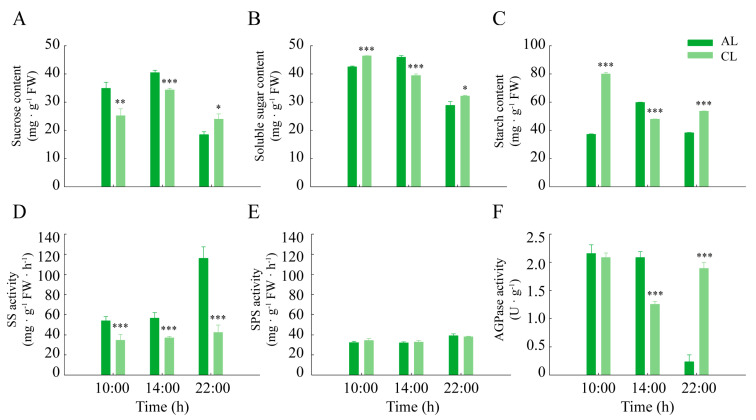
The carbohydrate metabolism level in ALs and CLs of *Camellia oleifera.* (**A**) Sucrose content, (**B**) total soluble sugar content, (**C**) starch content, (**D**) sucrose synthetase (SS) activity, (**E**) sucrose phosphate synthetase (SPS) activity, (**F**) ADP-glucose pyrophosphorylase (AGPase) activity. ALs: annual leaves; CLs: current-year leaves. * *p* < 0.05; ** *p* < 0.01; *** *p* < 0.001.

**Figure 6 plants-12-02161-f006:**
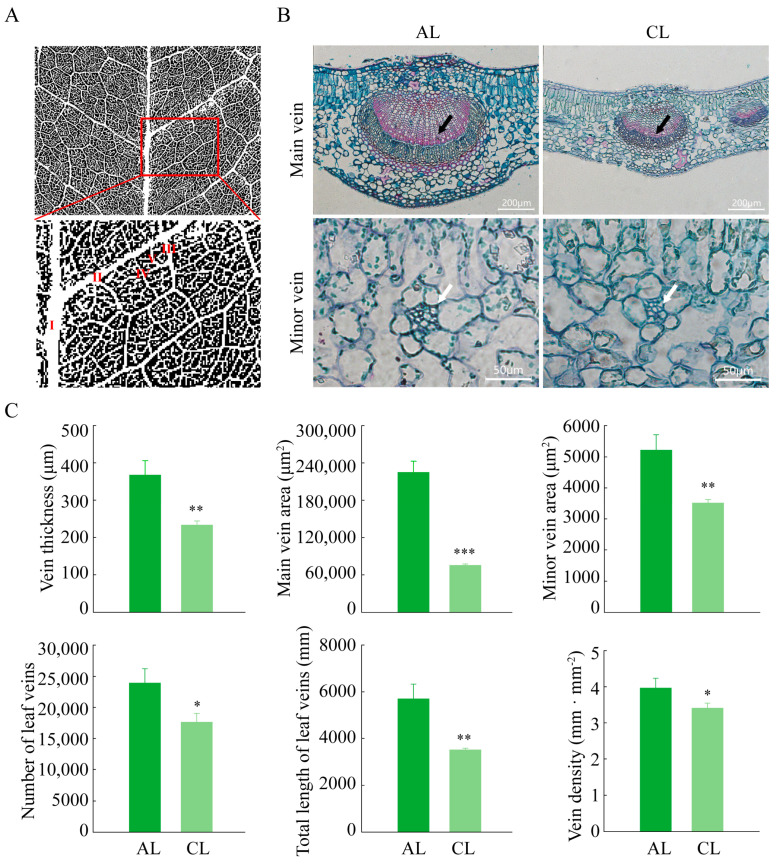
Leaf vein skeleton, microstructure and quantitative results in ALs and CLs of *Camellia oleifera*. (**A**) The image of the leaf vein skeleton processed using LEAF GUI software. The enlarged image in the red box shows a schematic diagram of vein grading from I to V, ranging from main veins to minor veins. (**B**) Paraffin section images of CLs and ALs. The black arrow points to the main vein, whereas the white arrow points to the minor vein. (**C**) Quantitative results of leaf vein structure characteristics in ALs and CLs. This includes leaf vein thickness (main vein), main vein area, minor vein area, number of leaf veins, number of leaf veins, total length of leaf veins and leaf vein density. CLs: current-year leaves; ALs: annual leaves. * *p* < 0.05; ** *p* < 0.01; *** *p* < 0.001.

**Figure 7 plants-12-02161-f007:**
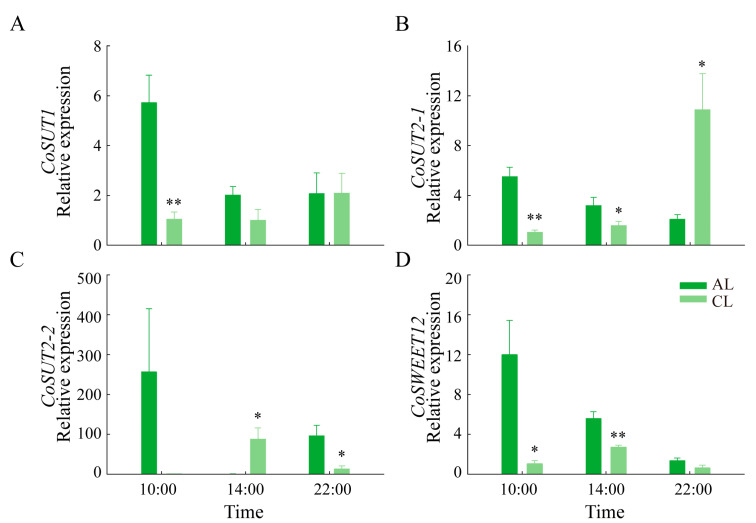
The expression levels of genes related to phloem loading in ALs and CLs of *Camellia oleifera*. (**A**–**C**) Relative expression of sucrose transporter (*SUT*) gene families. (**D**) Relative expression of Sugar Will Eventually be Exported Transporter (*SWEET*) gene families. CLs: current-year leaves; ALs: annual leaves. * *p* < 0.05; ** *p* < 0.01.

**Figure 8 plants-12-02161-f008:**
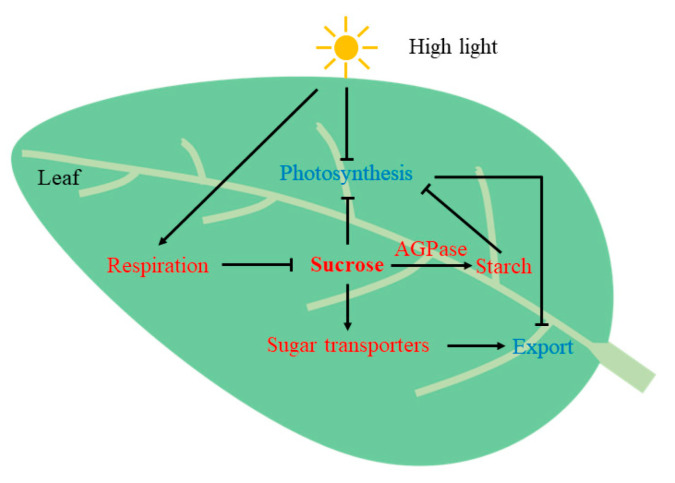
Suggested model for regulation of leaf photosynthesis related to changes in assimilate accumulation under high light in *Camellia oleifera.* The excessive accumulation of sucrose in leaves under high light can lead to a decrease in photosynthesis, which in turn inhibits the rate of assimilate export. The increase in leaf respiration rate and starch synthesis consumes sucrose to some extent. In addition, the leaves promote the transport of assimilates by enhancing the expression of the genes related to sugar transporters. Red and blue highlighted substances or processes indicate increased or decreased levels in the afternoon compared with the morning. Arrows show activation; bars show inhibition.

**Figure 9 plants-12-02161-f009:**
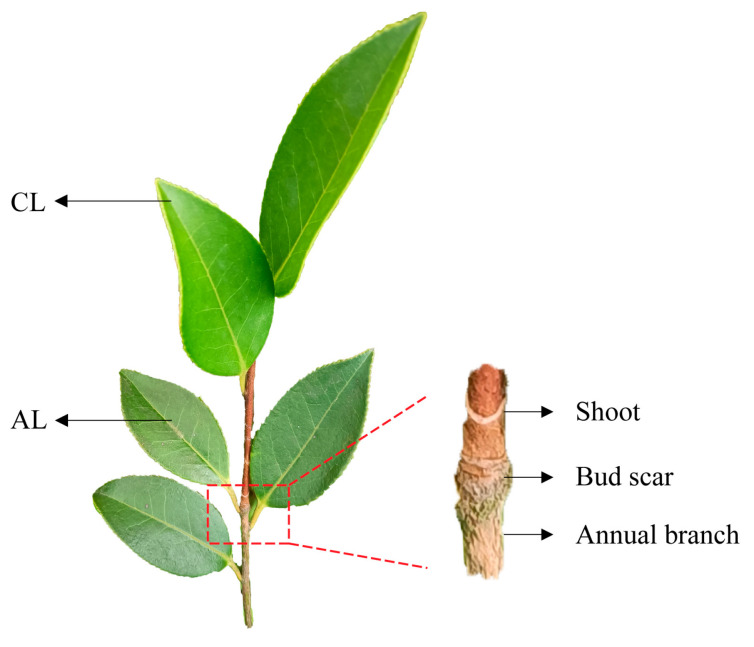
Relative positions of ALs and CLs in *Camellia oleifera*. The red box shows the bud scale marks between the new shoot and the annual branch, which are generated after the top bud scales of the sprouting new shoot fall off. ALs: annual leaves; CLs: current-year leaves.

**Table 1 plants-12-02161-t001:** Primers used for quantitative real−time PCR.

Target Gene	Primer Sequence
*CoACTR3*-F	GGTTGTAGTGGATGTTGGAGATGG
*CoACTR3*-R	AGCTGCTGGATGAAGAGAGTAAC
*CoSUT1*-F	ATGATAGTCCAACCCCTAGTCGG
*CoSUT1*-R	CCAGAACCCGACTACGAAAACC
*CoSUT2*-*1*-F	TGTGCGAATCTCAAGACCTG
*CoSUT2*-*1*-R	TGCCAACAATAGGTACCAC
*CoSUT2*-*2*-F	CAGATGGTGGTGTCAGTAGCAAG
*CoSUT2*-*2*-R	CAGCAATAGCCATGGGAACTTTG
*CoSWEET12*-F	CCAAAGAAAGAGAAGGGCAAGATCC
*CoSWEET12*-R	GCGTACATGATGATGGAGAAGATGG

## Data Availability

Data is contained within the article.

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
