# Peer review of "Diurnal Regulation of Leaf Photosynthesis Is Related to Leaf-Age-Dependent Changes in Assimilate Accumulation in Camellia oleifera"

_plants, 2023, doi:10.3390/plants12112161_

Round 1

Reviewer 1 Report

This paper reported diurnal changes in photosynthesis, chl fluorescence, sugar/starch content, carbon export rate, leaf anatomical structure, and expression of genes relating to sugar transport, in young (CL) and old (AL) leaves of Camellia oleifera on sunny days. This paper is very well-written, concise, and easy to follow, presented interesting results with sound discussions/conclusion. Therefore, this paper is suitable to be accepted for publications in Plants after some minor revisions as follows:

1. I think the gene expression results (possibly also leaf anatomy) are interesting and help substantiate discussions relating to sugar partitioning, and should be included in the main text instead of supplementary.

2. For Materials and Methods – (2.1) the authors should indicate the number of sunny days the measurements were undertaken, and the number of plants/number of leaves per plant investigated. (2.2) please explain the concept of obtaining the C-export data

3. For Results - when the authors compared, between CL and AL, the levels of decrease or increase in some parameters, they should show the percentage decrease or increase for clearer presentation.

4. Minor point – (1) the author name should be added at first mention of the plant i.e. ‘Camellia oleifera Abel.’ (2) the typing of ‘oleifera’ was not consistent (capital O was used in many places).

5. Title – ‘Camellia Oleifera’ – small letter ‘o’ should be used not ‘O’.

Reviewer 2 Report

1.      The abstract poorly written even did not cover the need of the study. This need to be revised with proper pattern and need.

2.      The introduction is incomplete according to the need of the study.  I would recommend a strong background for the study, and last paragraph of the study should explain the rational of the study, what was the need and importance of the study, 2ndly what was the rational, what authors want to answered, I mean what question were raised before and after the completion of the study. These thing are very necessary to explained nu the authors otherwise such introduction is incomplete.

3.      Material methods are incomplete having no such proper citations and poorly written. I strongly recommend the whole manuscript needs to revised with native English speaker and needs to be checked for the information.

4.      All the figures caption should be detail explaining the main theme what is happening in each figures. One-line caption are no more justifiable or acceptable.

5.      I would strongly recommend to analyses all the data statistically by an expert and do lettering on all graphs and table as well.

6.      Results need to check for grammar check. Also I would recommend to perform statistical analyses for all the data presented. ANOVA and LSD should be carry out.

7.      Discussion is poorly written and needs to be updated with valid logical statements and citation.

8.      Conclusion is the most important part and should explained the question raised before the study, and what questions are left after study which needs to be further analyze. 

Needs to be check with native english speaker
